# QUERY-EFFICIENT PLANNING WITH LANGUAGE MODELS

## ABSTRACT

Planning in complex environments requires an agent to efficiently query a world model to find a feasible sequence of actions from start to goal. Recent work has shown that Large Language Models (LLMs), with their rich prior knowledge and reasoning capabilities, can potentially help with planning by searching over promising states and adapting to feedback from the world. In this paper, we propose and study two fundamentally competing frameworks that leverage LLMs for query-efficient planning. The first uses LLMs as a *heuristic* within a search-based planner to select promising nodes to expand and propose promising actions. The second uses LLMs as a *generative planner* to propose an entire sequence of actions, query the world model, and adapt based on feedback. We show that while both approaches improve upon comparable baselines, using an LLM as a generative planner results in significantly fewer interactions. Our key finding is that the LLM as a planner can more rapidly *adapt* its planning strategies based on immediate feedback than LLM as a heuristic. We present evaluations and ablations on Robotouille and PDDL planning benchmarks and discuss connections to existing theory on query-efficient planning algorithms.

## 1 INTRODUCTION

Planning is the process of determining a sequence of feasible or optimal actions that guide an agent from an initial state to a desired goal state (LaValle, 2006). Planning assumes access to a world model, enabling the agent to simulate and evaluate potential actions without relying on trial-and-error in the real environment. However, in many domains, such as robot task and motion planning, *querying the world model is the most computationally expensive step* (Kaelbling & Lozano-Pérez, 2013; Garrett et al., 2021). For instance, each query involves running physics or geometric computations or even running a local optimizer. Consequently, planning algorithms must judiciously query the world model, relying on learning-based approaches to efficiently infer the most promising paths with minimal queries (Choudhury et al., 2018; Ichter et al., 2017; Khodeir et al., 2023).

Large language models (LLMs), trained on Internet-scale data, offer multiple opportunities to enable query-efficient planning. Notably, LLMs come with key capabilities such as *(1) powerful priors* to identify promising states that make progress toward the goal (Ahn et al., 2022), *(2) tractable posteriors* by easily conditioning on feedback to adaptively choose actions (Lee et al., 2023), and *(3) generating complex sequences* of actions to plan to the goal (Janner et al., 2021). Recent works leverage one or more such capabilities to design LLM-based agents that solve various decision-making tasks (Yao et al., 2022; Shinn et al., 2023b; Huang et al., 2022b; Zhao et al., 2023). However, we show that naively extending such LLM agents to the planning setting becomes quickly intractable. It must not only select among all possible state-action queries but condition on the history of all queries and observations.

Instead, one tractable way is to use a *LLM as a heuristic* within an existing planner. Heuristics guide a search tree from start to goal by selecting promising nodes to expand (Pearl, 1984). The planner provides the LLM with a restrictive set of nodes to choose from, making the problem more tractable for the LLM. This is the defacto approach that several recent works adopt to design LLM heuristics for classic breadth-first search (BFS) / depth-first search (DFS) (Yao et al., 2024) or for more advanced Monte Carlo tree search (MCTS) (Zhao et al., 2023; Hao et al., 2023b).

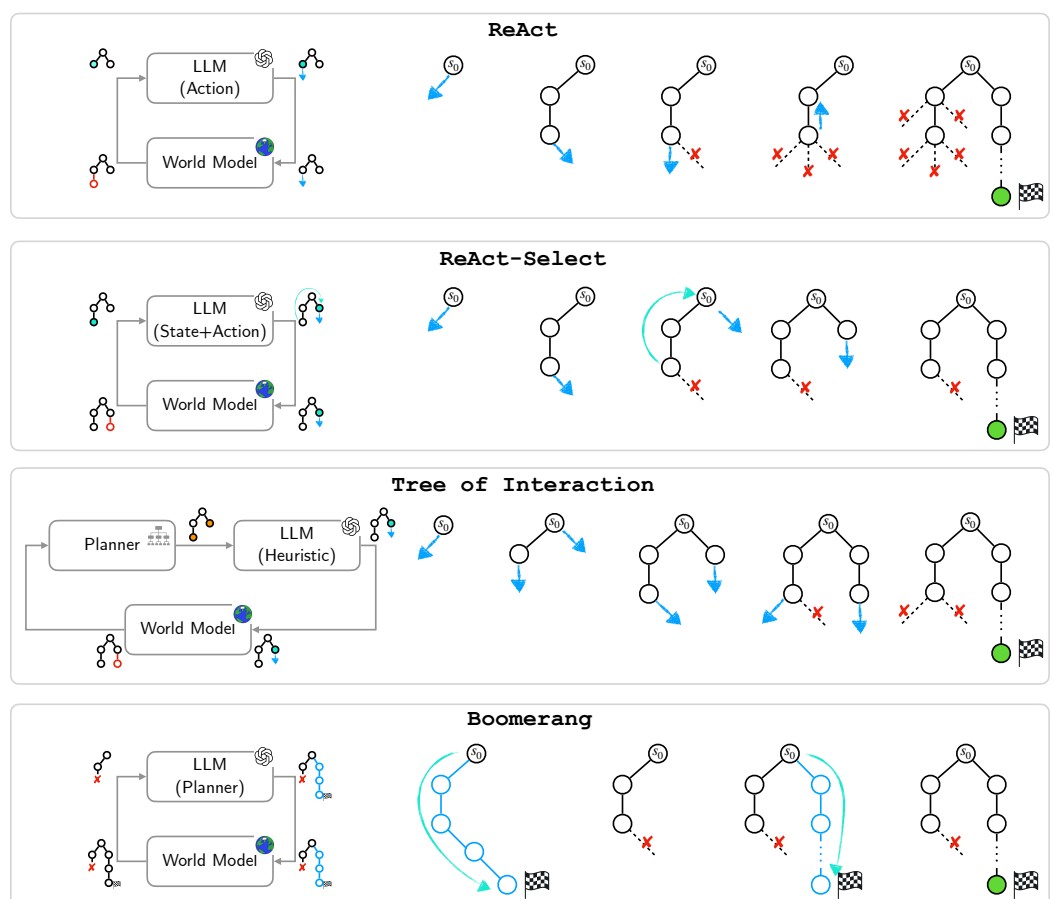

Figure 1: Overview of LLM planning methods that find a feasible path with minimal queries to a world model. ReAct selects actions only and must backtrack to undo its actions and take another path. ReAct-Select selects both states and actions, allowing it to immediately teleport to better states. Tree of Interaction (ToI) uses a planner to drive the search while using an LLM as a heuristic to select states. Boomerang generates an entire plan, allowing it to immediately switch to a new plan.

An alternative approach is to use a *LLM as a generative planner*. In this paradigm, the LLM directly generates a sequence of actions to the goal, and checks actions against a world model. If the plan is infeasible, the LLM conditions on this feedback to generate a new plan. This directly leverages the capability of transformers to predict entire sequences (Valmeekam et al., 2023; Pallagani et al., 2022; Lehnert et al., 2024), removing the need for an external planner. By reasoning over a set of promising paths rather than state-action queries, the decision space reduces as well. This idea is closely tied to a well-established framework of "lazy" search in classical planning literature, which has provable guarantees on finding shortest paths with minimal world model queries (Dellin & Srinivasa, 2016b; Mandalika et al., 2019; Hou et al., 2020).

We compare both paradigms on a series of fundamental planning tasks. ***Our key finding is that a LLM as a generative planner is more query efficient than planners using a LLM as a heuristic***. The key reason for this is that a LLM planner is *more adaptive to feedback from the world model* than a traditional planner using a LLM merely as a heuristic. For example, if the LLM planner discovers a cul-de-sac, its next plan can be in an entirely different basin to circumvent the cul-de-sac. On the other hand, the LLM heuristic is restricted to only choosing nodes offered by the planner and can continue selecting among nodes in the cul-de-sac without being able to change the search direction.

To study this problem, we propose two new algorithms (Tree of Interaction (ToI) and Boomerang) and repurpose two existing baselines (ReAct and ReAct-Select) for query-efficient planning (Fig. 1). Tree of Interaction (ToI) is an interactive version of prior

work, Tree of Thought (Yao et al., 2024), where the LLM is used as a heuristic within a BFS and DFS planner that interactively queries a world model. `Boomerang` is an interactive LLM planner that outputs action sequences from an initial state to the goal, and replans based on external feedback. We evaluate our methods on multiple planning domains proposed in the PlanBench (Valmeekam et al., 2024) benchmark, the Logistics domain, Grippers domain, and on Robotouille (Wang et al., 2023), a robotics simulator for cooking tasks. Our key contributions are:

1. Framework for query-efficient planning using LLMs.
2. Two new algorithms: `Tree of Interaction (ToI)` that uses LLM as a heuristic and `Boomerang` that uses LLM as a generative planner.
3. Evaluation of LLM and classical planners' query efficiency across PlanBench, Logistics, Grippers, and the Robotouille simulator. We show that `Boomerang` achieves the highest success rates, with 78% on Blocksworld, 82% on Logistics, 89% on Grippers, and 57% on Robotouille

## 2 PROBLEM FORMULATION

We are interested in planning problems where querying the environment world model is computationally expensive or resource intensive. This is a common assumption in many applications, especially in robotics. Planning robot motion in high-dimensional configuration spaces requires queries to computationally expensive collision checks (Hauser, 2015; Dellin & Srinivasa, 2016b; Mandalika et al., 2019; Hou et al., 2020). In the task and motion planning (TAMP) domain (Kaelbling & Lozano-Pérez, 2013; Ding et al., 2023; Lozano-Pérez & Kaelbling, 2014; Srivastava et al., 2014; Toussaint et al., 2018), each query is a high-level action proposed by a task planner, and the world model invokes an expensive motion planning subroutine to generate the next state. Traditional TAMP planners can take up to minutes to solve complex TAMP problems (Lin et al., 2023). Hence, real-time planning involves strategically selecting queries that *minimize* the total number of queries to the world model to find a feasible plan.

**Query-Efficient Planning as Sequential Decision Making.** Consider an agent operating within a known state space, $S$, and action space, $A$. We assume the existence of a deterministic world model, $M : S \times A \to S'$, which maps a state-action pair to the subsequent state in the world. The goal of the planner is to find a sequence of actions that joins the initial state $s_0$ and the goal state $s_g$, i.e., $\{s_0, a_0, s_1, a_1, \ldots, s_g\}$, where each transition $s_{i+1} = M(s_i, a_i)$ has been verified to be feasible by the world model. We can formulate the problem of query-efficient planning in this setting as one of sequential decision-making. At each decision-making step, the planner queries the world with a state-action pair, $q = \{s, a\}$. The world model responds with $r = \{s', e\}$ containing the next state $s'$ and an optional error message $e$ if the action is invalid.

**Leveraging LLMs for Query-Efficient Planning.** LLM agents have shown promising results in various sequential decision-making problems (Yao et al., 2022; Shinn et al., 2023b; Huang et al., 2022b). LLMs encode a vast array of commonsense priors that can be used to generate plausible plans from the outset. Good queries reveal useful information about the planning problem, which can be incorporated into the LLM agent's context to update its posterior for future decision-making. Formally, we represent the LLM agent's $k^{th}$ interaction with the world model as a policy $\pi(q_k|\phi, H_k)$, where $\phi$ is the world context that describes the problem domain in natural language, and $H_k = \{q_1, r_1, q_2, r_2, \ldots, q_{k-1}, r_{k-1}\}$, is the history of the past queries and responses. This formulation allows the LLM to leverage its pre-trained knowledge and update its priors with every interaction with the world model.

`ReAct` (Yao et al., 2022)-style prompting provides a simple recipe to use LLMs for query-efficient planning. The policy is represented as $\pi(a_k|\phi, H_k, s_k)$, where the last state $s_k$ is the result of the agent's last query to the world model. At every query step, the agent considers its history of interactions to generate reasoning for its decision before taking an action from its current state. However, `ReAct` policies are susceptible to getting trapped in local optima or *cul-de-sacs*. In such cases, the agent tries to backtrack from its current state instead of querying from more promising states in its history.

Representing the LLM as the more expressive policy $\pi(q_k|\phi, H_k)$, we can create `ReAct-Select`, a natural extension of `ReAct`. Unlike `ReAct`, which only uses the last state the agent ended up in for future queries, `ReAct-Select` allows the agent to decide both the state and the action for querying the world model. This flexibility allows `ReAct-Select` to revisit previously explored

states strategically and choose more efficient actions, establishing it as the gold-standard algorithm for optimal performance. However, the decision space of `ReAct-Select` is very large, i.e., $|Q| = |S| \times |A|$, choosing among all possible states and subsequent actions. In practice, we observe that `ReAct-Select` exhibits a recency bias where it degenerates to `ReAct` and stays committed to recent states as seen in Sec. 4.2.3.

## 3 APPROACH

We tackle the problem of query-efficient planning with LLMs, looking at algorithms that interactively query a world model. We previously discussed that the gold-standard algorithm, `ReAct-Select` (Sec. 2), is intractable as it requires conditioning on a large context containing the entire history of prior world model interactions while choosing queries from an enormous decision space.

Instead, we explore two fundamentally different approaches that constrain the decision space, yielding more tractable solutions. The first approach integrates LLMs into a heuristic search framework (Pearl, 1984). Rather than using the entire interaction history to determine the next query, the LLM serves as a heuristic within a higher-level search algorithm, ranking the most promising states in the search tree and guiding the selection of optimal actions. The second approach employs the framework of lazy search (Dellin & Srinivasa, 2016a), utilizing the LLM as a *generative planner*. In this method, the LLM generates an entire action sequence from start to goal based on its current understanding of the environment's world model. After each planning iteration, the LLM updates its internal world model using feedback from the environment. An illustration of both approaches is shown in Fig. 1.

---

**Algorithm 1** `ToI-BFS`

**Input:** Initial State $s_0$, Problem Description $\phi$, LLM Action Proposal $\pi_\theta$, LLM State Evaluator $V_\theta$, World Model $M$, Step Limit $T$, Actions to propose $k$, Best Candidates $b$
**Output:** Verified Plan $\{s_0, a_0 \dots s_g\}$
$S_0 \leftarrow \{s_0\}$
**for** $t$ in $1 \dots T$ **do**
    $\tilde{S}_t \leftarrow \{\}$
    **for** $s \in S_{t-1}$ **do**
        `// Propose Actions to Goal`
        $\tilde{A} \leftarrow \pi_\theta(s, \phi, k)$
        `// Query World Model for States`
        $\tilde{S}_t \leftarrow \tilde{S}_t \cup \{M(s,a) | a \in \tilde{A}\}$
        **if** `ReachedGoal`$(\tilde{S}_t)$ **then**
            **Return** `BacktrackPath()`
        **end if**
    **end for**
    $S_t \leftarrow$ `UpdateBeam`$(\{S_{t-1} \cup \tilde{S}_t\}, V_\theta, b)$
**end for**
**Return** $\{\}$

---

**Algorithm 2** `Boomerang`

**Input:** Initial State $s_0$, Problem Description $\phi$, LLM Generative Planner $P_\theta$, World Model $M$, Step Limit $T$
**Output:** Verified Plan $\{s_0, a_0 \dots s_g\}$
`// Initialize LLM History`
$H_0 \leftarrow \{\}$
**for** $t$ in $1 \dots T$ **do**
    `// Generate Plan using History`
    $\Pi \leftarrow P_\theta(s_0, \phi, H_{t-1})$
    `// Verify Plan by Querying World Model for State-Action Trajectory`
    $(\xi, \text{error}) \leftarrow M(s_0, \Pi)$
    **if** `ReachedGoal`$(\xi)$ **then**
        **Return** $\xi$
    **end if**
    `// Update LLM Context with Trajectory and Error Message`
    $H_t \leftarrow H_{t-1} \cup (\xi, \text{error})$
**end for**
**Return** $\{\}$

---

### 3.1 LLM AS A HEURISTIC: TREE OF INTERACTION (TOI)

`Tree of Interaction (ToI)` utilizes LLMs as a heuristic within an external planner. The planner maintains a search tree and invokes the LLM to choose which states to expand and what actions to propose. By judiciously choosing which states to expand, the LLM minimizes unnecessary queries to the world model to guide the search tree towards the goal. We build on prior work Tree of Thought (Yao et al., 2024), by incorporating queries to an external world model during the expansions phase. At a high level, the LLM is used to define two modules: *Action Proposal* and *State Evaluation*.

**Action Proposal** $\pi_\theta(\tilde{A} \mid s, \phi, k)$**.** This module proposes diverse actions to expand promising new states. An LLM is prompted to generate an action set, $\tilde{A}$, with $k$ actions from state $s$.

**State Evaluation** $V_\theta(s, \phi)$**.** This module evaluates states based on their potential to progress towards the goal. We utilize an LLM $V_\theta$ conditioned on a state $s$ and the problem context $\phi$. The state is classified into one of three categories based on how likely it is to reach the goal: *Impossible*, *Maybe*, and *Certain*. These rankings are used to guide the search tree towards better states.

Heuristic search algorithms can be constructed by combining the above modules in different ways. In particular, we describe one such algorithm, `ToI-BFS` (Alg. 1) (see Appendix A.6 for an algorithm of `ToI-DFS`). The algorithm uses beam-search to build a search tree greedily with a breadth-first search. Starting from some initial state $s_0$, the search attempts to expand states for $T$ iterations until the goal state $s_g$ has been expanded. A beam of size $b$ maintains the best candidates throughout the search. At each iteration, the *action proposer* module is first called to generate action-set $\tilde{A}$ of size $k$ for each state on the beam. Then, the world model is queried to produce the set of next states $\tilde{S}_t$ using each candidate state and its proposed action set. Finally, the *state evaluation* module updates the set of $b$ candidate states maintained in the beam search.

### 3.2 LLM as a Generative Planner: Boomerang

Instead of restricting the use of LLM agents within an external planner, `Boomerang` uses LLM as a generative planner that adapts to feedback from the world model. The LLM planner proposes a sequence of actions from the start to the goal state based on its *internal world model's* understanding of the problem domain. Then, it receives feedback from the true world model, which in turn updates the internal world model. Alg. 2 provides an overview of `Boomerang`. Key components are:

**Planning with an Internal World Model.** Equipped with context of the problem description and history of interactions with the world model, LLM agents can be prompted with Chain-of-Thought (Wei et al., 2022) techniques to build an *internal world model* of the problem domain. The agent uses this internal world model to reason and generate a plan $\Pi$, a sequence of actions that attempt to reach the goal from the start state. In every iteration, the generated plan receives feedback from the *true world model*, which is used by the LLM agent to propose new plans.

**Updating Internal World Model with Feedback.** At every iteration, the generated plan is validated by the *true world model* by rolling out actions from the start state using its generated plan. We re-use any queries made to the world model in previous iterations during this verification. If the true world model verifies that the plan reaches the goal state, the algorithm terminates and returns the verified trajectory. Otherwise, the true world model responds with a partial trajectory $\xi$ to the goal with an error message $e$ at some action in the plan. The partial trajectory and error message are appended into the LLM's prompt context, updating its *internal world model* for future iterations of planning.

We also note the connection of `Boomerang` with the framework of lazy search (Dellin et al., 2016) in motion planning in Appendix A.2. We derive a Bayesian regret bound that is sub-linear with the planning iterations needed by `Boomerang` before it returns a feasible solution. The core principle of laziness is to query edges that belong to promising paths to the goal state, thus minimizing queries to the world model to find either the shortest path (Dellin et al., 2016), a feasible path (Choudhury et al., 2017) or anytime path (Hou et al., 2020). Concretely, we can view the LLM as the policy $\pi(\xi|\phi_k)$, sampling plans $\xi$ from the posterior $P(\phi_k|\phi_{\text{prior}}, H_k)$ using feedback from the world model. Posterior sampling is a provable way (Hou et al., 2020) to tradeoff exploration and exploitation. While these classical works rely on discretization approaches to construct the posterior, we leverage the flexibility of the LLM in approximating posteriors. We conjecture that the empirical success of `Boomerang` is explained by this close connection.

Prompts for both algorithms (and variants) are provided in Appendix A.4 and A.5.

## 4 Experiments

### 4.1 Experimental Setup

**Planning Domains.** We evaluate both classical and LLM planners across a variety of fundamental planning problems. First, we assess all methods on the Blocksworld benchmark from Plan-Bench (Valmeekam et al., 2024), which consists of 600 block-rearrangement problems described in PDDL. Blocksworld has long been a classic AI planning benchmark and is now the de facto standard for evaluating LLMs' commonsense reasoning abilities in planning tasks. In addition, we create 100 planning problems in the Logistics and Grippers PDDL environments. We use PDDLGym (Silver & Chitnis, 2020) to interact with these environments, serving as the world model oracles. Beyond

these classic PDDL environments, we introduce 100 planning problems in the realistic robot cooking simulator, Robotouille Wang et al. (2023), which poses unique challenges due to complex task dependencies, including time delays and task multi-threading. Since Robotouille is not described in PDDL, we use the simulator itself as the world model oracle.

**Metrics.** We evaluate the efficiency of the planners in solving planning problems under a fixed budget of World Model Queries (WMQs). A *success* is defined as a planner finding a feasible path to the goal without exceeding the query budget. We also measure the average number of queries each method makes to the world model across all problems. Additionally, we introduce an *optimality* metric, which indicates whether a planner finds an optimal path within the WMQ budget. Beyond these planning metrics, we report the number of LLM API calls and input tokens used by the LLM-based methods to provide insights into the cost and runtime of the experiments

**Baselines.** We test a range of LLM planner approaches in our experiments. In the simplest case, we evaluate two *non-interactive* direct input-output methods that do not involve back-and-forth communication with the world model. `I/O` (Huang et al., 2022a) takes a problem description and an in-context demonstration as input, then generates a sequence of actions. `I/O + CoT` (Wei et al., 2022) builds on this by incorporating a chain of thought component. *Interactive* LLM planners are divided into two categories. The first is *heuristic* planners, including `ToI-DFS` and `ToI-BFS`. These planners embed the LLM as a heuristic within a higher-level classical search algorithm. We also evaluate *generative* planners that generate action sequences while interacting with the world model. `ReAct` (Yao et al., 2022) takes one-step actions within the environment, continuously adapting its strategy based on feedback from each step. `Boomerang` generates entire action sequences toward the goal before each interaction with the world model. For all PDDL environments, we also run state-of-the-art classical PDDL planners using the FastDownward system (Helmert, 2006). We conduct a hyperparameter sweep across multiple classical planner configurations and report the best-performing results as `Classical` (more details in A.10).

## 4.2 RESULTS AND ANALYSIS

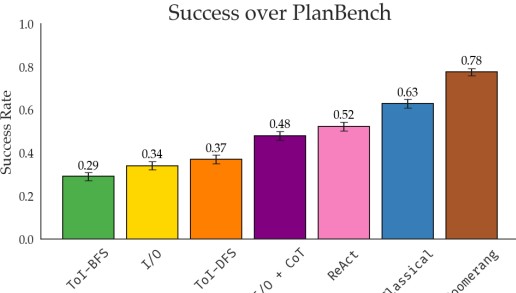

| | **LLM Calls** | **Token Usage** |
|---|---|---|
| I/O | 1.00 | 583.92 |
| I/O + CoT | 1.00 | 1,069.28 |
| ReAct | 13.69 | 51,262.80 |
| ToI-BFS | 39.58 | 32,163.30 |
| ToI-DFS | 28.32 | 23,454.00 |
| Boomerang | 5.69 | 38,235.73 |

Figure 2: Success of approaches that efficiently reached the goal within 20 world model queries. The classical planner is the best from Appendix A.10.

Table 1: Average LLM Calls and Token Usage per Blocksworld problem

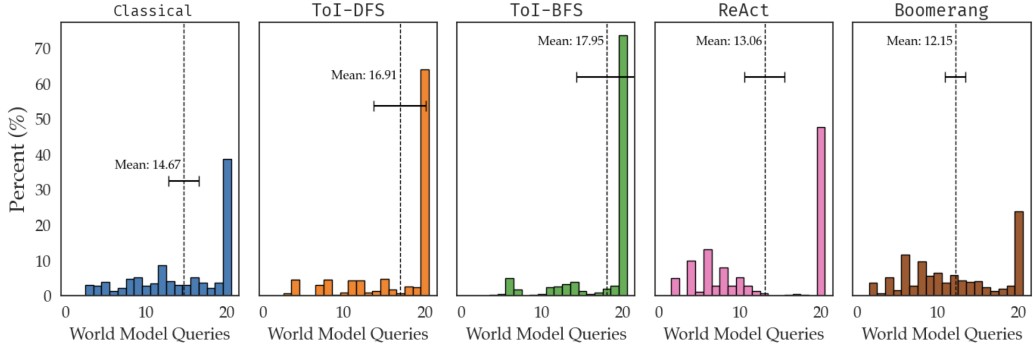

Figure 3: Histogram of interactive approaches' world model queries on Blocksworld problems. Count represents the number of runs that made a specific number of queries (total of 600 runs). Failures are capped at 20 world model queries.

### 4.2.1 OVERALL RESULTS

- `Boomerang` achieves 78% efficient success on 600 Blocksworld problems from PlanBench compared to `Classical` with 63% and `ReAct` with 52%. (Figure 2 and Sec 4.2.2).

- `Boomerang` achieves 82%, 89% and 57% efficient success on Logistics, Grippers, and Robotouille respectively compared to `ToI-DFS` with 4%, 31%, and 17% respectively and `Classical` with 5% and 13% on Logistics and Grippers. (Table 5 and Sec 4.2.2).

- `Boomerang` can overcome *cul-de-sacs* while `ReAct` and `ReAct-Select` struggle. See Sec 4.2.3.

- `I/O + P` and `I/O + CoT + P` surpass `ReAct` by 12.3% and 14.8% respectively after simple prompt changes. See Sec 4.2.4.

### 4.2.2 COMPARISON OF QUERY-EFFICIENCY, SOLUTION QUALITY, AND TOKEN USAGE

**Question 1.** *How query-efficient are the various approaches on the PlanBench dataset?*

Fig. 2 shows the overall success rates of all algorithms on 600 PlanBench problems with world model queries capped at 20. Fig. 3 shows a histogram of queries for all interactive planning approaches.

Among the LLM-based approaches, `Boomerang` has the highest success rate (0.78) and the lowest mean queries 12.15. `ReAct`, has a success rate of 0.52 with 13.06 mean queries; we scarcely observe successes for higher world model queries because `ReAct` tends to fail on longer horizon problems as it gets suck in *cul-de-sacs* and cycles between states due to misunderstanding the environment. We supplemented `ReAct` with a "reset" mechanism akin to Shinn et al. (2023a) to alleviate this, and observed slight improvements (see Appendix A.12 for results). `ToI-DFS` and `ToI-BFS` have the lowest success rates of 0.37 and 0.29 respectively, with mean expansions of 16.91 and 17.95 respectively; this is because these approaches select states without history which causes useful information for reaching the goal to be discarded in the next iteration of state selection.

The classical planner `Classical` has the second highest success rate of 0.63. This is attributed to its best-first search strategy and landmark-cut heuristic which performs efficiently on this domain (see Appendix 3). However, it still ends up querying the world model more (14.67).

**Question 2.** *How does the solution quality of various approaches compare on PlanBench?*

Fig. 4 shows the overall optimality of all algorithms on 600 PlanBench problems with world model queries capped at 20.

Among the LLM-based approaches, `Boomerang` has the highest optimality rate (0.69), despite no optimality guarantees. This is because resetting to the start after collecting feedback allows optimal shooting towards the goal. `ReAct` has an optimality rate of 0.47; this shows that `ReAct` can incorporate feedback well if it does not get stuck in *cul-de-sacs* along the way. Finally, `ToI-DFS` and `ToI-BFS` have the lowest optimality rates

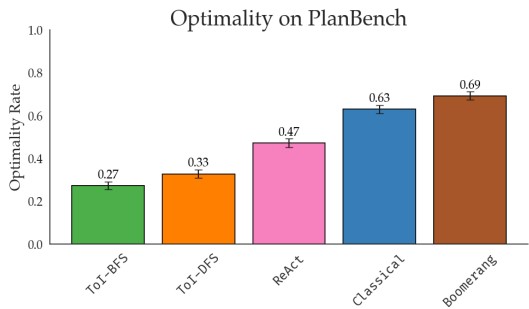

Figure 4: Optimality rate of interactive approaches that reach the goal within 20 world model queries

of 0.33 and 0.27. This is because in shorter-horizon problems these methods can afford to query the world model for various paths to the goal.

The classical planner `Classical` has an optimality rate of 0.63 which matches its success rate in Fig. 2. `Classical` uses best-first-search which implies its heuristic does not underestimate the true cost, guaranteeing it finds an optimal path.

The optimality of LLM could be increased using LPG Gerevini et al. (2011) if important; however, this would incur more world model queries which we are trying to minimize.

**Question 3.** *How token-efficient are the various LLM approaches?*

Table 1 contains the average number of LLM queries and tokens used per run. The cheapest methods are `I/O` and `I/O + CoT` which are queried once for an entire action sequence. `I/O + CoT` has

more token usage ($1k$ vs $0.5k$) since it contains state prediction reasoning in its in-context examples. The next group of cheapest methods includes `ToI-BFS` and `ToI-DFS` which make the most number of queries (39.58 and 28.32) although have medium token usage (~$32k$ and ~$23k$). `ToI-BFS` is more expensive since it maintains twice as many candidates which can each expand actions. Finally, the most expensive methods are `Boomerang` and `ReAct`. Even though these have lower queries (5.69 and 13.69) they have higher token usage (~$38k$ and ~$51k$). These methods both include history in their input but `ReAct` contains more since it needs to take various actions towards the goal while `Boomerang` shoots various action sequences towards the goal, keeping its history relatively shorter. See Appendix A.11 for the total costs for all LLM-based approaches.

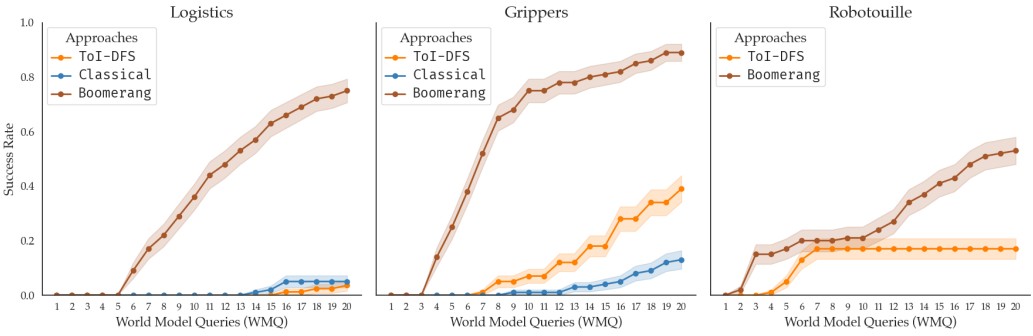

Figure 5: Comparison of `ToI-DFS`, `Boomerang`, and `Classical` planners on Logistics, Grippers, and Robotouille: We chart the success rate given various world model query budgets and observed that `Boomerang` is most query-efficient at reaching the goal. The applicable classical planners are the best from Appendix A.10.

.

**Question 4.** *How do the various approaches perform on other domains?*

Figure 5 contains comparions of `ToI-DFS`, `Classical`, and `Boomerang` on 100 examples each of Logistics, Grippers, and Robotouille. We specifically compare these methods on additional domains since `ToI-DFS`, `Classical`, and `Boomerang` represent the best instantiations of LLM heuristic, classical, and LLM generative planner methods. In all 3 datasets `Boomerang` is by far the best at reaching the goal across all world model query budgets. `ToI-DFS` is also more query efficient than `Classical` in Grippers; this is due to the large action space in the domain which classical planners naively expand due to their lack of real world grounding which LLMs can exploit.

### 4.2.3 Analysis of Failure Modes of Approaches

**Question 5.** *What are failure modes for `ReAct` and `ReAct-Select`?*

We look at a qualitative example where all interactive approaches fail except `Boomerang`. This instance has an optimal plan length of 12 and involves rearranging a stack of blocks into another stack. `ReAct` makes quick progress towards the goal (red block at the bottom) in Fig. 6; however, it ends up stuck in a cul-de-sac – it picks up the red block thinking it can be put underneath other blocks and puts it back down, creating an endless cycle. This is counter-intuitive since `ReAct` keeps everything in history but in practice we observe that `ReAct` selectively pays attention to the most recent history and tends to ignore past important signals in favor of newer ones. Similarly, in Fig. 7, `ReAct-Select` begins its search by selecting the next state, and as the history fills with this reasoning the only state that is ever chosen is the next state. `ReAct-Select` degenerates to `ReAct` and inherits all of its issues. `Boomerang` can combat this issue by outputting action sequences, keeping its history relatively shorter, and resetting to the start state, which allows it to escape *cul-de-sacs* as shown in Fig. 8. For `ToI-BFS` and `ToI-DFS` failures, see Appendix A.7.

### 4.2.4 Ablations for Better LLM Plan Generation

**Question 6.** *How far can we improve non-interactive methods for planning?*

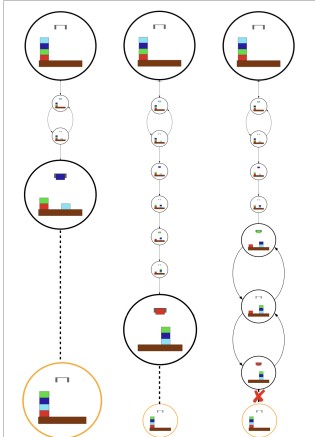
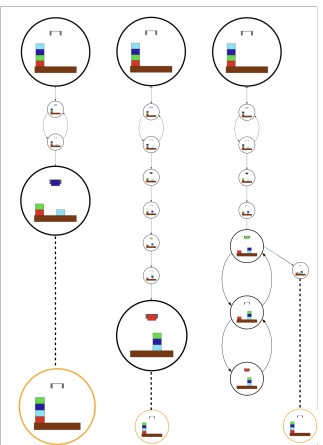
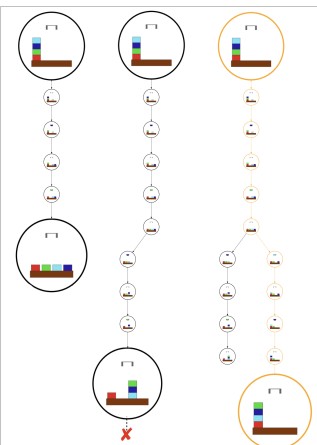

Figure 6: Timelapse of a failure where `ReAct` repeatedly enters a cul-de-sac when attempting to backtrack (goal in orange)

Figure 7: Timelapse of a failure where `ReAct-Select` repeatedly selects the next state and falls into a similar failure to `ReAct` (goal in orange)

Figure 8: Timelapse of a success where `Boomerang` enters a cul-de-sac but corrects its trajectory after resetting to the start (goal path in orange)

We break down the success and failure modes of `I/O` and `I/O + CoT` in Fig. 9. We make a simple prompt change to the action space of `I/O` resulting in `I/O + P` and additionally output the goal repeatedly in `I/O + CoT + P` resulting in `I/O + CoT + P` (see Appendix A.8 for our design choices). Most notably, `I/O + P` and `I/O + CoT + P` achieve higher success than `ReAct` (by 12.3% and 14.8%). The key reason for this is because the `I/O + P` variants have the most useful information immediately available while `ReAct` has useful information scattered through its exploration history. Similarly, `Boomerang` benefits from its shorter history since it can immediately act upon useful information in its next decision; however, some improvements can still be made. Future work investigating how to extract the most useful information and keep that in context can immensely reduce history length and improve decision-making.

## 5 RELATED WORKS

To motivate our approach and differentiate from other works, we present related work that uses LLMs as heuristics, incorporate feedback into LLMs, and uses LLMs for planning in PDDL environments.

**LLMs as Heuristics.** Existing work suggest that LLMs struggle with end-to-end planning (Valmeekam et al., 2024), but may be effective as components as planners (Kambhampati et al., 2024). One format of this paradigm is using LLMs as heuristics for search (Yao et al., 2023; Hao et al., 2023a; Zhao et al., 2023; Xie et al., 2023; Lu et al., 2021), with an external planner. One related work we build off is Tree-of-Thought (ToT) (Yao et al., 2023). This approach builds off Chain-of-Thought (CoT) (Wei et al., 2022), where an LLM outputs step-by-step reasoning before a final response. ToT uses an LLM to generate 'thoughts' and an external

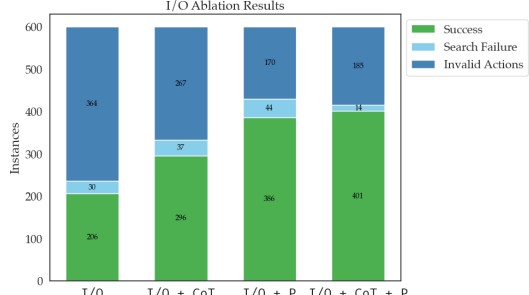

Figure 9: Stacked bar plot of successes and failures modes across `I/O` variants. 'Invalid Actions' refers to an outputted action sequence that contains an invalid action. 'Search Failure' refers to a valid outputted action sequence that does not reach the goal.

planner to explore the thought space; the LLM then acts as a heuristic by selecting the next best

thoughts to expand on. Other works like RAP (Hao et al., 2023a) and LLM-MCTS (Zhao et al., 2023), use Monte Carlo Tree Search where an LLM guides the simulations when expanding a state.

While search infrastructure makes planning with LLMs more robust, this is overly complex for simple problems. We observed LLMs exhibited some abilities of planning and searching independently, and when supplemented with environment feedback could mostly operate as end-to-end planners.

**Planning with Environment Feedback.** Another key ingredient to planning effectively with LLMs is incorporating environment feedback (Yao et al., 2022; Shinn et al., 2023b; Madaan et al., 2023; Sun et al., 2023; Gou et al., 2024; Huang et al., 2022b). A simple example of this paradigm is ReAct (Yao et al., 2022). ReAct iteratively builds a trajectory with environment feedback following each step. Reflexion (Shinn et al., 2023b) adds to ReAct by building multiple ReAct trajectories with a reflection step in between them that remarks on learnings from failed trajectories, improving subsequent ones. Decision-Pretrained Transformer (DPT) (Lee et al., 2023) connects transformers trained on state-action-reward tuples to Bayesian posterior sampling. We can generally view prompting with environment feedback in a similar light: we view an LLM to be a model with parameters partially "learned" by feedback in its prompt. This formulation is interesting since posterior sampling has well studied regret bounds (Lee et al., 2023; Osband et al., 2013).

**LLMs and PDDL.** Works such as PlanBench (Valmeekam et al., 2024) and Valmeekam et al. (2023) have tested prompting GPT models on PDDL domains on whole plan generation; findings show an inability to track states and evaluate state transitions. Others have explored tackling PDDL problems though the lens of coding. Silver et al. (2023) describes a prompting approach to generate Pythonic programs given a domain definition. Plansformer (Pallagani et al., 2022) is a CodeT5 model (Wang et al., 2021) fine-tuned to generate full plans. The model was exceptional on all tested domains (Blocksworld, Hanoi, Grippers, and Driverlog) and exhibited some signs of transfer learning across domains. Some have excluded LLMs from the planning phase entirely (Liu et al., 2023; Guan et al., 2023; Lyu et al., 2023). For example, LLM+P (Liu et al., 2023) converts a natural language description of a planning task into a PDDL domain and instance, queries a traditional planner, then converts the planner output back to natural language.

## 6 DISCUSSION

In this paper, we look at query-efficient planning with language models. We propose two approaches, `Tree of Interaction (ToI)` where an LLM is used as a heuristic and `Boomerang` where an LLM is used as a generative planner. We evaluate our approach on PDDL domains and show that `Boomerang` is more query-efficient than both classical and LLM planning baselines. Key to this is `Boomerang` adapting the entire plan based on feedback from the world model, which has close ties to known results on posterior sampling for query-efficient planning. Two interesting future directions:

**Scaling to longer horizons.** As the horizon increases, `Boomerang` may fail to generate good plans. A path to scaling would be to make it plan with its internal world models using any number of approaches (Yao et al., 2023; Besta et al., 2023; Zhao et al., 2023), query the world model to validate the plan, and iterate with feedback.

**Scaling to complex planning problems requiring geometric reasoning.** Our end goal is to solve complex task and motion planning problems. While LLMs can reason about semantics but struggle to reason about grids (Lehnert et al., 2024) or geometry (Trinh et al., 2024). An exciting direction of future work is to look at using LLM to plan at a higher semantic level, pass this to a low-level geometric planner to produce actions, and crucially adapt the high-level planner based on failures of the geometric planner to guide it better.

## 7 LIMITATIONS

While `Boomerang` is promising, it is hampered during prolonged searches. We frequently observed the method reproposing failed action sequences during long horizon examples, which suggests a tendency to forget about feedback early on in its context. The `ToI-BFS` and `ToI-DFS` methods are also overly dependent on the LLM providing quality rankings of states; erroneous rankings cause states to be unnecessarily explored in `ToI-BFS` or can send a search on the path to a dead end in `ToI-DFS`. Additionally, the LLM heuristic itself also ranks states independently from one another which tend to make the heuristic inadmissible (failing to guarantee optimal paths) and inconsistent (making it possible to select visited nodes). Finally, all proposed methods can be costly and slow to run, requiring multiple calls to an LLM.

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

## A    APPENDIX / SUPPLEMENTAL MATERIAL

### A.1    BROADER IMPACT

`Tree of Interaction (ToI)` and `Boomerang` are planning approaches that incorporate external world feedback to boost LLMs' planning abilities. As these approaches improve, they can be applied in unstructured environments like software applications that search the Web or robots that assist us in our homes to improve the quality of our lives. On the other hand, giving full autonomous control to these LLM planners can lead to dangerous actions or advice; it is important to apply additional safety checks on generated plans and for future work to better align LLMs toward safe plans.

### A.2    ANALYZING THE PERFORMANCE OF BOOMERANG

**Why is laziness essential for query efficiency?** Assume we have a perfect heuristic $h^*(s, s_g)$. Let's say the heuristic is used by A* search. A* will expand only the vertices corresponding to the optimal path $\xi^*$. However, at every vertex expansion step, A* will evaluate every outgoing edge from the vertex, i.e. degree $k$. Hence, the total number of queries will be $k|\xi^*|$, where $|\xi^*|$ is the number of edges in the optimal path. Contrast this to a lazy shortest path Dellin & Srinivasa (2016b), which initializes an internal model where every edge is assumed to be feasible, finds the shortest path in its internal model, and then queries the path to update its model. Initializing the model with the true model would yield the true shortest path in the first iteration $\xi^*$, resulting in $|\xi^*|$ queries. Hence, even with access to perfect information, A* heuristic search is $k$ times more query expensive than lazy shortest path. We refer the reader to Mandalika et al. (2018) for a more rigorous proof, that shows lazy shortest path is strictly more query efficient than A*.

**Connection between `Boomerang` and Posterior Sampling** While lazy shortest path Dellin & Srinivasa (2016b) is query-efficient, it does not leverage any prior information about the world model. We now establish a connection between Boomerang and posterior sampling for reinforcement learning (PSRL) Osband et al. (2013). This connection enables us to translate Bayesian regret bounds from posterior sampling to the expected number of queries made by Boomerang.

Let $\phi$ be the initial context provided to the LLM, that describes the problem domain in natural language. Let $P_\theta(M^\star|\phi)$ be the prior over MDPs that the LLM $\theta$ implicitly models based on its prior knowledge. Similarly let $P_\theta(M^\star|\phi, H_t)$ be the posterior over MDPs given history $H_t = \{q_1, r_1, q_2, r_2, \ldots, \}$ of query/response pairs. At every iteration $t$, Boomerang samples a model $M_t \sim P_\theta(M^\star|\phi, H_t)$ in its reasoning step, which is then used to generate an optimal plan $\xi_t$. Every state, action, and next-state $(s, a, s') \in \xi_t$ is then queried to the world model and the history $H_{t+1}$ is updated accordingly. Once a feasible path is found, the game terminates. We define a reward function $R^M(s, a, s') = 0$ if $(s, a, s')$ is feasible, else $R^M(s, a, s') = -1$ if $(s, a, s')$ is not feasible. Note that the true reward $R^{M^\star}(s, a, s')$ is unknown to the agent, and must be discovered through queries. We denote $V_\xi^M = \sum_{s,a,s' \in \xi} R(s, a, s')$ as the cumulative reward of a path $\xi$ in model $M$. In other words, the value is the number of infeasible transitions.

We define the regret at every round $t$ as $\Delta_t = V_{\xi_t}^{M^\star} - V_{\xi^*}^{M^\star}$, where $\xi^*$ is the optimal path in the true model $M^\star$. In other words, the regret measures the number of infeasible transitions of the path $\xi_t$ (since the second term $V_{\xi^*}^{M^\star}$ is always 0). Finally, the Bayesian Regret is the expected total cumulative regret, i.e. $\textsc{BayesRegret}(\text{T}) = \mathbb{E}\sum_{t=0}^{T}\Delta_t$. A bound on the Bayesian Regret is a bound on the expected number of infeasible transitions queried until a feasible path is found. A query-efficient algorithm has a low Bayesian Regret. Having defined a mapping between our problem and PSRL, we adapt the Bayesian Regret bound for PSRL to provide a bound for Boomerang.

**Theorem 1** *Let $S$ be the number of states, $A$ be the number of actions, and $\tau$ be the length of the longest path. Boomerang, after $T$ iterations, has a bounded Bayes Regret of* $\textsc{BayesRegret}(\text{T}) \leq \mathcal{O}(\tau\sqrt{SAT}\log(SAT))$*, which bounds the number of infeasible edges queried till a feasible path is found.*

The proof of the theorem above follows from Osband et al. (2013).

### A.3 MODELS AND HYPERPARAMETERS

All approaches use `gpt-4-turbo`. The interactive LLM approaches are allowed to make 20 decisions for exploring the environment. `ToI-DFS` and `ToI-BFS` both use $k = 2$ and `ToI-BFS` uses $b = 2$. We experimented with alternative values for $b$, but found little change in success *with a fixed query budget*: on 10 randomly sampled Blocksworld problems, there were 3, 2, and 2 successes for $b$ values 2, 3, and 5, respectively. For a fixed length action sequence, increasing either parameter entails increasing the number of queries. So while the "branching factor" of the search is wider, the search can not traverse enough successive actions to reach the goal. Although we only experimented with varying $b$ values, we believe modifying $k$ entails a similar effect. Finally, all approaches use a temperature of 0.7 and use no in-context examples unless otherwise specified. We empirically observed no difference in performance on `Boomerang` when additionally supplied with in-context examples as shown in Table 2 below

| Example | Domain | |
|:---:|:---:|:---:|
| | Blocksworld | Gripper |
| None | 8/10 | 6/10 |
| Blocksworld | 8/10 | 6/10 |
| Gripper | 8/10 | 6/10 |

Table 2: We vary the domain of the in-context example (including providing no in-context example) and the test example and observe no difference in performance on 10 randomly sampled Blocksworld and Grippers problems with `Boomerang`.

### A.4 PROMPTS

#### A.4.1 STATE TRANSLATION PROMPT

Below is a sample prompt and `gpt-4-turbo` output for state translation. We find that `gpt-4-turbo` is able to effectively translate state in the form of PDDL predicates into a natural language form. Our prompt can be flexible across domains: the parts in "Below is a description of the environment:" and "The actions are formatted as follows:" can easily be swapped out depending on the domain

```
You are an assistant that summarizes PDDL predicates into natural
↪ language.

You will receive the following to summarize into natural language
Predicates: ...
Objects: ...

where
- 'Predicates' is a list of PDDL predicates
```

```
810   - 'Objects' is a list of objects
811
812   You may also receive the following:
813   Goal: ...
814
815   where
816   - 'Goal' is a list of PDDL predicates
817
818   It is important to incorporate all predicates and objects into the
819   ↪  succinct summary.
      An agent will be using this summary to understand the state of the
820   ↪  environment and
      if anything is missing, it may lead to confusion.
821
822
823   Below is a description of the environment:
824   The 4-operator version of the classic Blocksworld. This domain
825   ↪ consists of a set of blocks, a table and a robot hand. The
826   ↪ blocks can be on top of other blocks or on the table; a block
827   ↪ that has nothing on it is clear; and the robot hand can hold
828   ↪ one block or be empty. The goal is to find a plan to move from
829   ↪ one configuration of blocks to another.
830
831   The actions are formatted as follows:
      - put-down(x:default) where x is the block to put down
832   - pick-up(x:default) where x is the block to pick up
833   - stack(x:default,y:default) where x is stacked on top of y
834   - unstack(x:default,y:default) where x is unstacked from the top
835   ↪ of y
836   --------- Provided Instance --------
837   Predicates: on(f:default,c:default), on(d:default,g:default), on(g
838   ↪ :default,b:default), ontable(a:default), clear(a:default), on(h:
839   ↪ default,e:default), clear(d:default), handempty(), clear(h:
840   ↪ default), ontable(e:default), on(b:default,f:default), ontable(
841   ↪ c:default)
842   Objects: h:default, a:default, g:default, e:default, c:default, d:
      ↪ default, f:default, b:default
843   ---------- GPT-4 response ----------
844   In the current state of the environment, there are eight blocks
845   ↪ labeled from A to H. The configuration is as follows:
846
847   - Block A is on the table and is clear.
848   - Block B is on top of Block F.
849   - Block C is on the table.
850   - Block D is on top of Block G and is clear.
851   - Block E is on the table.
852   - Block F is on top of Block C.
853   - Block G is on top of Block B.
854   - Block H is on top of Block E and is clear.
855
```

### A.4.2 GOAL TRANSLATION PROMPT

Since goals in PDDL are also given in terms of predicates, we also use GPT4 to translate goals into a natural language form. Below is an example from Blocksworld.

```
860   You are an assistant that summarizes PDDL predicates into natural
861   ↪ language.
862
863   You will receive the following to summarize into natural language
      Predicates: ...
```

```
Objects: ...

where
- 'Predicates' is a list of PDDL predicates
- 'Objects' is a list of objects

You may also receive the following:
Goal: ...

where
- 'Goal' is a list of PDDL predicates

It is important to incorporate all predicates and objects into the
↪  succinct summary.
An agent will be using this summary to understand the state of the
↪  environment and
if anything is missing, it may lead to confusion.

Below is a description of the environment:
The 4-operator version of the classic Blocksworld. This domain
↪ consists of a set of blocks, a table and a robot hand. The
↪ blocks can be on top of other blocks or on the table; a block
↪ that has nothing on it is clear; and the robot hand can hold
↪ one block or be empty. The goal is to find a plan to move from
↪ one configuration of blocks to another.

The actions are formatted as follows:
- put-down(x:default) where x is the block to put down
- pick-up(x:default) where x is the block to pick up
- stack(x:default,y:default) where x is stacked on top of y
- unstack(x:default,y:default) where x is unstacked from the top
↪ of y
--------- Provided Instance --------
Goal: on(a:default,g:default), on(b:default,f:default), on(e:
↪ default,a:default), on(h:default,e:default)
---------- GPT-4 response ----------
The goal in this environment is to arrange a specific
↪ configuration of blocks. Specifically, the objectives are to
↪ have block A on top of block G, block B on top of block F,
↪ block E on top of block A, and block H on top of block E. This
↪ configuration must be achieved by strategically moving the
↪ blocks using the robot hand through actions such as picking up,
↪  putting down, stacking, and unstacking.
```

### A.4.3  BOOMERANG ACTION PROPOSAL PROMPT

Below is a sample prompt and GPT4 output for the action proposal component of `Boomerang`.
Here, the LLM is tasked with generating a trajectory from the start state to the goal. This builds off
the start state and goal translation examples from before (they are provided as input in the action
proposal prompt).

```
You must propose a sequence of actions given previous interactions
↪  with the environment
from the starting state to the goal state.

You will receive the initial state and the goal as follows:
Optional[Error Feedback: ...]
States Visited: ...
<action1>: ...
```

```
<action2>: ...
...
<actionN>: ...
Starting State: ...
Valid Actions: ...
Goal State: ...

where
- 'States Visited' are the states you visited in your previous
↪ action sequence
  - This will be empty if this is your first action sequence
  - Each action will be followed by the state that resulted from
  ↪ executing that action
- 'Starting State' is the state you will start from
- 'Valid Actions' are the actions you can take in the starting
↪ state
- 'Goal State' is the state you need to reach to achieve the goal
- 'Error Feedback' includes feedback about either
  - the sequence of actions you proposed in the previous step
  ↪ included an invalid action
  - the sequence of actions you proposed in the previous step did
  ↪ not reach the goal state

Always format your response as follows:
Reflect: ...
Think: ...
Action Sequence: <action1>, <action2>, ..., <actionN>

where:
- 'Reflect' includes lessons learned about the rules of the
↪ environment
  - Upon receiving error feedback, reflect on the feedback and
  ↪ propose a new plan
    - If the action is invalid, first verify that the action isn't
    ↪ malformed
      - Refer to the action format in the environment description
    - If it isn't malformed, consider why the action is invalid at
    ↪ that state
  - Consider which action(s) in the previous sequence led to the
  ↪ error
- 'Think' includes your thought process for the next action
↪ sequence to propose
  - Use your current and previous reflections to inform the next
  ↪ action sequence
- 'Action Sequence' is the sequence of actions you propose to take
↪  in the environment from the starting state to the goal state
  - This sequence should be a comma-separated list of actions
  - Each action should be formatted to match the templates in the
  ↪ environment description.

Note that the action sequence must always start from the 'Starting
↪  State' and end at the 'Goal State'.

Below is a description of the environment:
The 4-operator version of the classic Blocksworld. This domain
↪ consists of a set of blocks, a table and a robot hand. The
↪ blocks can be on top of other blocks or on the table; a block
↪ that has nothing on it is clear; and the robot hand can hold
```

```
↪ one block or be empty. The goal is to find a plan to move from
↪ one configuration of blocks to another.

The actions are formatted as follows:
- put-down(x:default) where x is the block to put down
- pick-up(x:default) where x is the block to pick up
- stack(x:default,y:default) where x is stacked on top of y
- unstack(x:default,y:default) where x is unstacked from the top
↪ of y
--------- Provided Instance --------
Starting State:
In the described environment, the hand of the robot is empty,
↪ indicating it's not currently holding any blocks. The blocks
↪ are arranged as follows:
- Blocks 'e', 'c', and 'a' are on the table.
- Blocks 'g' is on block 'b', and block 'd' is on block 'g',
↪ forming a vertical stack.
- Block 'f' is on block 'c', and block 'b' is on block 'f', also
↪ forming another stack.
- Block 'h' is on block 'e'.

Blocks 'a', 'd', and 'h' are clear, meaning no other blocks are
↪ stacked on top of them. This setup includes eight blocks
↪ labeled from 'a' to 'h'.
Valid Actions:
- pick-up(a:default)
- unstack(d:default,g:default)
- unstack(h:default,e:default)
Goal State:
The goal in this environment is to arrange a set of blocks in a
↪ specific configuration where block A is on block G, block B is
↪ on block F, block E is on block A, and block H is on block E.
---------- GPT-4 response ----------
Reflect:
This is the first attempt at creating an action plan in this
↪ environment. There are no previous actions or errors to learn
↪ from at this point. The goal is to rearrange blocks to achieve
↪ a specific configuration, and the valid actions at this point
↪ involve picking up or unstacking the clear blocks.

Think:
To reach the goal state, several moves must be made taking into
↪ account the structure and limitations of the current block
↪ arrangement:
1. Blocks 'a', 'd', and 'h' are clear and can be moved. However,
↪ to achieve the goal state, block 'a' needs to be placed on
↪ block 'g' and block 'h' on block 'e', which itself needs to be
↪ on block 'a'. Therefore, block 'a' should initially remain on
↪ the table.
2. Block 'd' must be unstacked from block 'g' and placed on the
↪ table.
3. Block 'h' should be unstacked from block 'e' and placed on the
↪ table.
4. 'e' should be picked up and placed on block 'a'.
5. Finally, block 'h' can be stacked on block 'e'.

Action Sequence:
unstack(d:default,g:default), put-down(d:default), unstack(h:
↪ default,e:default), put-down(h:default), pick-up(e:default),
```

```
1026  ↪ stack(e:default,a:default), pick-up(h:default), stack(h:default,
1027  ↪ e:default)
1028
```

1029

### A.4.4 ToI-BFS AND ToI-DFS ACTION PROPOSAL PROMPT

We again use the state and goal translation examples from before. Below is the action proposal prompt for ToI-BFS and ToI-DFS. The prompt tasks the LLM to pick several actions from the set of available actions (so they may be expanded later).

```
You will propose various options for actions that could be taken
↪ in the environment to make progress towards the goal.

You will receive the initial state and the goal as follows:
Optional[Error Feedback: ...]
Number of Actions: ...
Current State: ...
Valid Actions: ...
Goal State: ...

where
- 'Number of Actions' is the number of actions you need to propose
- 'Current State' is the state you will start from
- 'Valid Actions' are the actions that can be executed in the
↪ current state
- 'Goal State' is the state you need to reach to achieve the goal
- 'Error Feedback' includes feedback about the actions you
↪ proposed in the previous step such as
  - the sequence of actions you proposed included an invalid
  ↪ action
  - the sequence of actions did not include the specified number
  ↪ of actions

Always format your response as follows:
Think: ...
Actions: <action1>, <action2>, ..., <actionN>

where:
- 'Think' includes reasoning about the actions you will propose to
↪  take
  - You should consider actions from the valid actions that will
  ↪ help you reach the goal state
- 'Actions' are <Number of Actions> actions that you propose to
↪ take at the current state
  - These actions should come directly from the valid actions
  - This sequence should be a comma-separated list of actions
  - The actions should be formatted exactly as they are in the
  ↪ environment description

Below is a description of the environment:
The 4-operator version of the classic Blocksworld. This domain
↪ consists of a set of blocks, a table and a robot hand. The
↪ blocks can be on top of other blocks or on the table; a block
↪ that has nothing on it is clear; and the robot hand can hold
↪ one block or be empty. The goal is to find a plan to move from
↪ one configuration of blocks to another.

The actions are formatted as follows:
- put-down(x:default) where x is the block to put down
- pick-up(x:default) where x is the block to pick up
```

```
1080    - stack(x:default,y:default) where x is stacked on top of y
1081    - unstack(x:default,y:default) where x is unstacked from the top
1082    ↪ of y
1083
1084    Number of Actions: 2
1085    Current State:
1086    In the current state of the environment, there are eight blocks
1087    ↪ labeled from A to H. The configuration is as follows:
1088
1089    - Block A is on the table and is clear.
1090    - Block B is on top of Block F.
1091    - Block C is on the table.
1092    - Block D is on top of Block G and is clear.
1093    - Block E is on the table.
1094    - Block F is on top of Block C.
1095    - Block G is on top of Block B.
1096    - Block H is on top of Block E and is clear.
        --------- Provided Instance --------
1097    Additionally, the robot hand is currently empty. This setup forms
1098    ↪ a complex structure of blocks arranged on the table and on top
1099    ↪ of each other, with several blocks being clear and directly
1100    ↪ accessible.
1101    Valid Actions:
1102    - pick-up(a:default)
1103    - unstack(d:default,g:default)
        - unstack(h:default,e:default)
1104    Goal State:
1105    The goal in this environment is to arrange a specific
1106    ↪ configuration of blocks. Specifically, the objectives are to
1107    ↪ have block A on top of block G, block B on top of block F,
1108    ↪ block E on top of block A, and block H on top of block E. This
1109    ↪ configuration must be achieved by strategically moving the
1110    ↪ blocks using the robot hand through actions such as picking up,
1111    ↪  putting down, stacking, and unstacking.
        ---------- GPT-4 response ----------
1112    Think: To progress towards the goal configuration, I need to start
1113    ↪  by making Block A available to be stacked on Block G. Given
1114    ↪ that Block A is clear and on the table, it can be easily picked
1115    ↪  up. The next step will be to make Block G accessible by
1116    ↪ unstacking Block D from it. The robot hand can then be free to
1117    ↪ move Block A once Block G is cleared.
1118
1119    Actions: pick-up(a:default), unstack(d:default,g:default)
1120
```

### A.4.5 ToI-BFS AND ToI-DFS STATE EVALUATION PROMPT

We follow the some initial state and goal translation examples from before. Below is the state evaluation prompt for ToI-BFS and ToI-DFS. The evaluation is necessary in deciding which states to explore later.

```
You will evaluate the current state based on its likelihood to be
↪ on the path to the goal state.

You will receive the initial state and the goal as follows:
Optional[Error Feedback: ...]
Current State: ...
Goal State: ...

where
- 'Current State' is the state you will evaluate
- 'Goal State' is the state you need to reach to achieve the goal
- 'Error Feedback' includes feedback about the evaluation you
↪ provided in the previous step such as
  - the evaluation you provided was not formatted correctly

Always format your response as follows:
Think: ...
Rating: <sure/maybe/impossible>

where:
- 'Think' includes reasoning about the likelihood of the current
↪ state being on the path to the goal state
  - You should consider the current state and the goal state
- 'Rating' is your evaluation of the current state based on the
↪ likelihood of it being on the path to the goal state
  - sure: the current state is definitely on the path to the goal
  ↪ state
  - maybe: the current state might be on the path to the goal
  ↪ state
  - impossible: the current state is definitely not on the path to
  ↪  the goal state

Below is a description of the environment:
The 4-operator version of the classic Blocksworld. This domain
↪ consists of a set of blocks, a table and a robot hand. The
↪ blocks can be on top of other blocks or on the table; a block
↪ that has nothing on it is clear; and the robot hand can hold
↪ one block or be empty. The goal is to find a plan to move from
↪ one configuration of blocks to another.

The actions are formatted as follows:
- put-down(x:default) where x is the block to put down
- pick-up(x:default) where x is the block to pick up
- stack(x:default,y:default) where x is stacked on top of y
- unstack(x:default,y:default) where x is unstacked from the top
↪ of y
--------- Provided Instance --------
Current State:
In the given Blocksworld environment, we have a configuration
↪ involving eight blocks labeled from 'a' to 'h'. The current
↪ state of these blocks is as follows:

- Block 'f' is on block 'c', which is on the table.
- Block 'd' is on block 'g', which itself is on block 'b'. Block '
↪ b' is then placed on block 'f'.
- Block 'h' is on block 'e', which is directly on the table.
- The robot hand is currently holding block 'a'.
```

```
- Blocks 'd' and 'h' are clear, meaning there is nothing on top of
↪  them.

This setup forms a couple of stacks with various blocks on the
↪ table and some blocks being clear. The robot's task involves
↪ manipulating these blocks possibly to rearrange them into a
↪ different configuration.
Goal State:
The goal in this environment is to arrange a specific
↪ configuration of blocks. Specifically, the objectives are to
↪ have block A on top of block G, block B on top of block F,
↪ block E on top of block A, and block H on top of block E. This
↪ configuration must be achieved by strategically moving the
↪ blocks using the robot hand through actions such as picking up,
↪  putting down, stacking, and unstacking.
---------- GPT-4 response ----------
Think: In the current state, the robot hand is holding block 'a',
↪ which is a positive start since one goal is to have block 'a'
↪ on block 'g'. However, block 'g' is currently under block 'd'
↪ and on top of block 'b', which complicates the immediate
↪ placement of 'a' on 'g'. The goal also includes having block 'b
↪ ' on block 'f', but currently, block 'f' is under block 'b' and
↪  carrying block 'c', so significant rearrangement is needed
↪ there. Block 'e' must end up with block 'h' on top, which
↪ matches the current state partially as 'h' is already on 'e'.
↪ However, 'e' must also be stacked on 'a' per the goal, which is
↪  not yet the case. The current configuration of blocks, despite
↪  being complex, is not entirely misaligned with the goal
↪ structure. Strategic moves like unstacking and rearranging are
↪ required, but the current state provides some foundational
↪ elements in line with the goal.

Rating: maybe
```

### A.5 CONTEXT

Our prompts are designed to allow for quickly injecting brief context information to align an LLM's priors for the current environment. We emphasize that the context is brief because the interactive approaches can fill in the gaps through interactions with the environment.

#### A.5.1 BLOCKSWORLD

```
The 4-operator version of the classic Blocksworld. This domain
↪ consists of a set of blocks, a table and a robot hand. The
↪ blocks can be on top of other blocks or on the table; a block
↪ that has nothing on it is clear; and the robot hand can hold
↪ one block or be empty. The goal is to find a plan to move from
↪ one configuration of blocks to another.

The actions are formatted as follows:
- put-down(x:default) where x is the block to put down
- pick-up(x:default) where x is the block to pick up
- stack(x:default,y:default) where x is stacked on top of y
- unstack(x:default,y:default) where x is unstacked from the top
↪ of y
```

#### A.5.2 GRIPPERS

```
 Given a robot with one or more gripper hands, transport a number
 ↪  of balls from their starting rooms to their destination rooms
 ↪ .

 Examples of how some actions might be formatted are as follows:
 - move(robot1,room1,room2) to move robot robot1 from room room1
 ↪ to room room2
 - pick(robot1,ball2,room3,gripper3) to have robot robot1 pick up
 ↪  ball ball2 using gripper gripper3 in room room3
 - drop(robot1,ball2,room3,gripper3) to have robot robot1 drop
 ↪ ball ball2 using gripper gripper3 in room room3
```

#### A.5.3 LOGISTICS

```
 Transport packages within cities via trucks, and between cities
 ↪ via airplanes. Locations within a city are directly connected
 ↪ (trucks can move between any two such locations), and so are
 ↪ the cities. In each city there is exactly one truck, each city
 ↪  has one location that serves as an airport.

 The actions are formatted as follows:
 - drive-truck(t0,l1-2,l3-2,c2) where t0 is a truck driving from
 ↪ location l1-2 to location l3-2 in city c2
 - fly-airplane(a0,l1-2,l3-4) where a0 is the airplane flying
 ↪ from the location l1-2 in city 2 to location l3-4 in city 4
 - load-airplane(p0,a1,l2-3) where p0 is the package loaded onto
 ↪ airplane a1 at location l2-3 in city 3
 - load-truck(p0,t1,l2-3) where p0 is the package loaded onto
 ↪ truck t1 at location l2-3 in city 3
 - unload-airplane(p0,a1,l2-3) where p0 is the package unloaded
 ↪ from airplane a1 at location l2-3 in city 3
 - unload-truck(p0,t1,l2-3) where p0 is the package unloaded from
 ↪  truck t1 at location l2-3 in city 3
```

## A.6 ToI-DFS Algorithm

---

**Algorithm 3** ToI-DFS

---

**Input:** Initial State $s_0$, Problem Description $\phi$, LLM Action Proposal $\pi_\theta$, LLM State Evaluator $V_\theta$, Value Threshold $v_{min}$, World Model $M$, Step Limit $T$, Actions to propose $k$
**Output:** Verified Plan $\{s_0, a_0 \dots s_g\}$
    `// Initialize DFS Queue of States`
    $S_Q \leftarrow \{s_0\}$
    **for** $t$ in $1 \dots T$ **do**
        `// Choose next state to expand`
        $s \leftarrow S_Q.\text{pop}()$
        `// Propose Actions to Goal`
        $\tilde{A} \leftarrow \pi_\theta(s, \phi, k)$
        `// Query World Model for States`
        $\tilde{S}_t \leftarrow \tilde{S}_t \cup \{M(s, a) | a \in \tilde{A}\}$
        `// Keep States above the Value Threshold`
        $S_Q \leftarrow S_Q \cup \{s | s \in \tilde{S}_t, V_\theta(s, \phi) > v_{min}\}$
        **if** `ReachedGoal`$(S_Q)$ **then**
            **Return** `BacktrackPath()`
        **end if**
    **end for**
    **Return** $\{\}$

---

We show ToI-DFS in Alg. 3. The algorithm builds a search tree greedily with a depth-first search and uses the heuristic as a termination condition. Starting from some initial state $s_0$, the search attempts to expand states for $T$ iterations until the goal state $s_g$ has been expanded. At each iteration, we pop a state off of the DFS queue $S_Q$. Then, the *action proposer* module is called to generate action-set $\tilde{A}$ of size $k$ for that state. Then, the world model is queried to produce the set of next states $\tilde{S}_t$ using the popped state and its proposed action set. Finally, the *state evaluation* module evaluates each state to add to the queue and ignores the states that are below the value threshold $v_{min}$.

## A.7 Tree of Interaction (ToI) Qualitative Failures

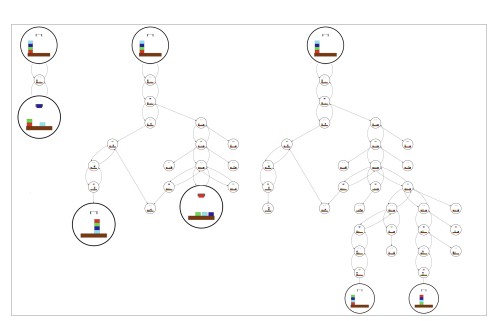
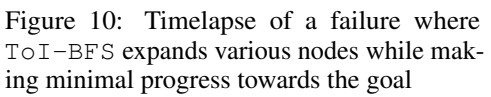

Figure 10: Timelapse of a failure where ToI-BFS expands various nodes while making minimal progress towards the goal

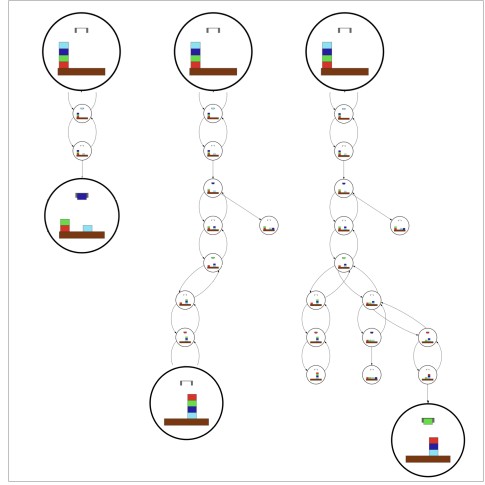

Figure 11: Timelapse of a failure where ToI-DFS expands various subtrees while making minimal progress towards the goal

Figure 10 shows ToI-BFS inefficiently expanding large numbers of nodes and Figure 11 shows ToI-DFS inefficiently expanding various subtrees. Both approaches make minimal progress to-

wards the goal because despite having queried vast information, the lack of history in `Tree of Interaction (ToI)` makes information useful only for expanding the next states.

### A.8 Prompting Ablation Details

To explore the generative power of LLMs, we run ablations on the `I/O` and `I/O + CoT` prompts to boost their performance. In `I/O` we found that the LLM misuses the 'pickup' action, intended for block-on-table interaction, in Blocksworld rather than the 'unstack' action, intended for block-on-block interaction, when picking up blocks from other blocks. We addressed this by enhancing 'pickup' and 'putdown' to allow for block-on-block interactions in the approach `I/O + P` where P stands for prompt engineering. For `I/O + CoT` we additionally found that the LLM benefits from tracking the environment state throughout its action sequence. Though not always accurate, this extra grounding reduced invalid action errors. Finally, we observed cases where the LLM hallucinated a goal midway through its generation (i.e. stating "I have reached the goal since the final state has: the orange block is on top of the blue block" when the goal was actually to have the orange block on top of the red one. We remedied some of these cases in `I/O + CoT + P` by instructing the goal condition to be repeated following every action, next state pair.

We break down the success and failure modes in Fig. 9. The 'Invalid Actions' failure mode refers to an outputted action sequence that contains an invalid action while 'Search Failure' refers to a valid outputted action sequence that does not reach the goal. Adding in prompt changes makes a significant difference - `I/O + P` achieves a 87% performance improvement over `I/O` and `I/O + CoT + P` achieves an 38% performance improvement over `I/O + CoT`. We note that `I/O + P` makes 53% fewer invalid action failures than `I/O`. `I/O + CoT + P` achieves 3.9% better performance than `I/O + P` but makes 8.8% more invalid action failures. This is attributed to small state prediction errors getting worse throughout a plan.

### A.9 Dataset Statistics

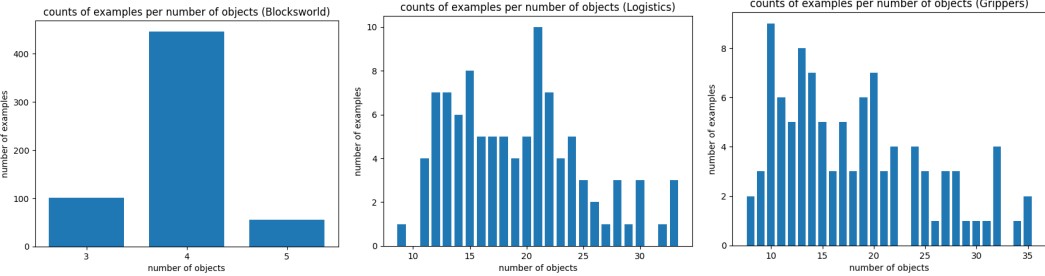

Figure 12: Amount of examples for different numbers of objects in Blocksworld, Logistics, and Grippers

### A.10 Comparisons of Classical Planners and Boomerang

| Heuristic | Search | | | | | | Boomerang | |
| --- | --- | --- | --- | --- | --- | --- | --- | --- |
| | A* | | Weighted A* (w=3) | | Best First Search | | | |
| | Success | Avg WMQ | Success | Avg WMQ | Success | Avg WMQ | Success | Avg WMQ |
| Landmark Cut | $0.628 \pm 0.196$ | 18.91 | $0.632 \pm 0.020$ | 18.04 | $0.628 \pm 0.020$ | **17.75** | - | - |
| Fast Forward | $0.547 \pm 0.020$ | 22.94 | $0.622 \pm 0.020$ | 19.75 | $0.628 \pm 0.020$ | 19.72 | - | - |
| Causal graph | $0.522 \pm 0.020$ | 26.58 | $0.513 \pm 0.020$ | 22.88 | $0.518 \pm 0.020$ | 22.65 | - | - |
| - | - | - | - | - | - | - | $0.85 \pm 0.015$ | **10.67** |

Table 3: Comparison of combinations of heuristic and search methods vs `Boomerang` on Blocksworld. We use the Best First Search and Landmark Cut combination for `Classical` in the results.

| Heuristic | Search | | | | | | Boomerang | |
|---|---|---|---|---|---|---|---|---|
| | A* | | Weighted A* (w=3) | | Best First Search | | | |
| | Success | Avg WMQ | Success | Avg WMQ | Success | Avg WMQ | Success | Avg WMQ |
| Landmark Cut | $0.05 \pm 0.022$ | 145.95 | $0.05 \pm 0.022$ | 107.26 | $0.05 \pm 0.022$ | 107.40 | - | - |
| Fast Forward | $0.05 \pm 0.022$ | 122.43 | $0.05 \pm 0.022$ | 97.79 | $0.05 \pm 0.022$ | 97.79 | - | - |
| Causal graph | $0.05 \pm 0.022$ | **81.98** | $0.05 \pm 0.022$ | **81.98** | $0.05 \pm 0.022$ | **81.98** | - | - |
| - | - | - | - | - | - | - | $0.82 \pm 0.038$ | **15.52** |

Table 4: Comparison of combinations of heuristic and search methods vs `Boomerang` on Logistics. We refer to any search strategy using the causal graph heuristic for `Classical` in the results.

| Heuristic | Search | | | | | | Boomerang | |
|---|---|---|---|---|---|---|---|---|
| | A* | | Weighted A* (w=3) | | Best First Search | | | |
| | Success | Avg WMQ | Success | Avg WMQ | Success | Avg WMQ | Success | Avg WMQ |
| Landmark Cut | $0.08 \pm 0.027$ | 358.17 | $0.08 \pm 0.027$ | 156.44 | $0.08 \pm 0.027$ | 157.78 | - | - |
| Fast Forward | $0.13 \pm 0.034$ | 121.53 | $0.13 \pm 0.034$ | **112.25** | $0.13 \pm 0.034$ | **112.25** | - | - |
| Causal graph | $0.07 \pm 0.026$ | 1681.64 | $0.07 \pm 0.026$ | 479.28 | $0.07 \pm 0.026$ | 476.30 | - | - |
| - | - | - | - | - | - | - | $0.89 \pm 0.031$ | **11.39** |

Table 5: Comparison of combinations of heuristic and search methods vs `Boomerang` on Grippers. We refer to either the Weighted A* or Best First Search strategies using the fast forward heuristic for `Classical` in the results.

### A.11   TOTAL COSTS OF LLM-BASED APPROACHES ON 600 BLOCKSWORLD EXAMPLES

| | I/O | I/O + CoT | ToI-BFS | ToI-DFS | ReAct | Boomerang |
|---|---|---|---|---|---|---|
| Total costs | \$3.63 | \$7.29 | \$192.98 | \$140.72 | \$307.58 | \$229.41 |

Table 6: Total API call costs for LLM-based approaches on 600 Blocksworld problems

### A.12   REACT, REFLEXION, AND BOOMERANG

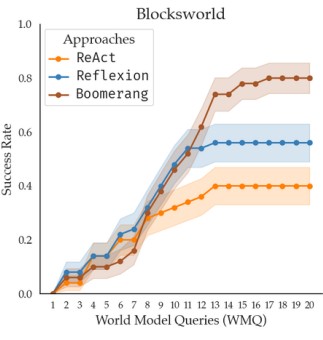

Figure 13: Comparison of Reflexion, , and `ReAct` on 100 randomly sampled examples of Blocksworld. We vary the world model query budget and observe Reflexion is able to outperform `ReAct` but not match `Boomerang`

.

Reflexion Shinn et al. (2023a) produces one action at a time similar to `ReAct`; however, similar to `Boomerang`, it can restart to the start whenever a mistake is made. We use this mechanism to address `ReAct` cycling states by prompting for feedback when such a cycle is detected. Specifically, we reset to the starting state and provide feedback to the LLM that cycling occurred for it to reflect on its mistake.

Reflexion performed better than `ReAct` as expected, but underperformed `Boomerang`. Interestingly, `Reflexion` tends to cycle after resetting from the start, similar to the failure modes discussed in

Section 4.2.3. While `Reflexion` essentially gets the same feedback that `Boomerang` does, we hypothesize that since the feedback is over a longer context window that it is harder for `Reflexion` to effectively incorporate this compared to `Boomerang` which outputs entire trajectories and can immediately reflect and act from a past mistake.

