# OpenReview forum: "Query-Efficient Planning with Language Models"
_ICLR.cc/2025/Conference — ICLR 2025 Conference Withdrawn Submission_

### Official Review · Reviewer_tztc · 2024-10-25

**Soundness:** 2
**Presentation:** 3
**Contribution:** 2
**Rating:** 5
**Confidence:** 4

**Summary:**

The paper studies two framework which uses LLM for planning. In the first framework, one uses a classical planning algorithm to search solutions whilst relying on an LLM as a heuristic to explore promising states and propose actions. The second is to use an LLM as a generative planner directly, proposing a sequence of actions (i.e., plan) and getting feedback from the world model.

The paper found that empirically, using LLM as a generative planner is more query efficient. That is, it uses fewer queries to the world model during the planning process to find solutions. The paper provides extensive experimental results to justify its claims. In addition, the paper shows various failure modes for different methods, showing that using LLM as a heuristics doesn't allow it to explore totally different states in cul-de-sacs situations, as compared to LLM as a generative planner, which can switch to entirely different plans straightaway.

**Strengths:**

1. The paper is easy to read, because the main idea is quite straightforward (using LLMs as a heuristic or generative planners are arguably known methods which have been used in prior papers).
2. The experiments are well conducted and extensive. They are broken down such that each question can be answered directly from a figure/table.
3. Failure modes are shown in the paper to corroborate each method's contribution and capabilities.

**Weaknesses:**

1. The main idea and motivation of the paper is quite simple. While this is not really an issue per se, I feel that there are several places in the paper where contributions are overclaimed. (a) The abstract mentioned that "we propose and study two fundamentally competing frameworks ...  the first uses LLMs as a heuristic within a search-based planner to select promising nodes to expand and propose promising actions ... the second uses LLMs as a generative planner". However, I am quite sure this paper is not the first to propose these two frameworks. For example, [1,2] all uses a LLM as a heuristic in one way of another to expand the search tree (or propose actions) and [3] uses LLMs to propose candidate plans. (b) the paper also claims to have introduced Tree of Interaction (TOI), which " ... maintains a search tree and invokes the LLM to choose which states to expand and what actions to propose". However, similar approaches have been proposed in prior works [1]. Therefore, I'd suggest the author making it clearer what the novel contribution of the paper is. Perhaps, the intention of the paper is to study the two frameworks from a query-efficient point of view?

2. In light of the point above, I feel the paper's proposed methods are not sufficiently novel. The paper's writing should reflect the novel parts clearly (for example, [1,2] have all used LLM as heuristic to expand the search tree, so can we really consider TOI, which does the same, a novel contribution?).

3. The paper claims that using LLM as a generative planner (Boomerang) needs to query the world model fewer times. In the boomerang setting, we would need to pass in the entire proposed sequence of actions (the entire plan) into the world model each time. On the other hand, for TOI, we need to simply pass in a single action at each planning stage. While the results show that Boomerang queries the model fewer times in total, at each stage the world model might incur larger computational cost because it needs to handle an entire sequence of actions. Shouldn't this be taken into account before discussing about query-efficiency? Would a metric such as time taken be better?

[1] Yu. et, al. Prompt-Based Monte-Carlo Tree Search for Goal-oriented Dialogue Policy Planning, 2023

[2] Hui. et, al. RoT: Enhancing Large Language Models with Reflection on Search Trees, 2024

[3] Kambhampati. et, al. LLMs Can't Plan, But Can Help Planning in LLM-Modulo Frameworks, 2024

**Questions:**

Line 403: There is random dot on the paper.

Line 373: Citation formatting is incorrect

Figure 2 & 4 seems to show the success rates of various algorithms on PlanBench. What is the difference between the two figures? Also, for some reason the same algorithm is giving different performance in both plots?

---

### Official Review · Reviewer_AQJT · 2024-11-02

**Soundness:** 3
**Presentation:** 4
**Contribution:** 3
**Rating:** 8
**Confidence:** 4

**Summary:**

In this paper, the authors use LLMs to help with planning problems in two competing methods.  The first (ToI) uses LLMs as a (dynamic) heuristic embedded within a traditional search-based planner to rank nodes and actions based on the goal, current state, and constraints.  The second method (Boomerang) uses LLMs as a generative planner to directly output a sequence of actions that take the agent from the initial state to the goal.  The plan is then checked with a world model, and, if needed, a  new plan is generated.  In this paper, “query-efficient” refers to querying the world model as this is often the most computationally expensive step in the planning process.

**Strengths:**

The paper is well-organized and very clearly written.  It gives the key finding right up front: LLM as a generative planner is more query efficient than planners using LLM as a heuristic. Because LLMs are more adaptive to feedback from the world model than a traditional planner using LLM as a heuristic.

The key contributions are clearly defined:
a.	Framework for query-efficient planning using LLMs
b.	Two new algorithms: ToI and Boomerang
c.	Evaluation of LLM and classical planners’ query efficiency

The analysis of the results is very good (with a couple of exceptions noted below).

Overall, this is a very good quality paper.

**Weaknesses:**

The main weakness is the omission of the question: the methods proposed use LLMs to improve query efficiency with a goal of reducing computational cost.  Is there a net benefit to computational cost when the increased cost of the LLM calls is considered?  (See "Questions" for more detail)

**Questions:**

MOST IMPORTANT QUESTION THAT SHOULD BE ASKED (but is not):  The paper proposes methods for improving (world model) query efficiency.  It uses LLMs to do this.  How does the increased computational cost of the LLM queries offset the computational cost savings by reducing the number of world model queries?  For typical problems is there actually a net reduction in computational cost? (Alternatively: for what types of problems can we expect a net reduction of computational cost, and for what types should we not expect a net reduction?)
a.	This question strikes me as more relevant than token usage or number of LLM calls (though these would be needed for this analysis)
b.	I realize this is problem dependent, but the authors should be able to calculate a comparison for the benchmark domains presented.
c.	In Table 1 and Figure 3 it looks like using Boomerang saves a (mean) ~2.5 world model calls over Classical (Fast Downward) on Blocksworld problems at the cost of (avg) ~5.69 LLM calls at an avg (38,000 tokens/call).  Is this a net computational cost reduction?

Additional questions:
1.  Were the LLM heuristics ever converted to natural language so we can understand the strategies they are using?  It would be very interesting to see examples of those strategies in natural language for both the Action Proposer and State Evaluator heuristics
2.  Even though the Boomerang algorithm was more query-efficient, are there potential benefits of the ToI algorithm if query efficiency isn’t the main goal?  ToI is an interesting algorithm but there is not much discussion of it's potential benefits.
3.  Line 118: What is meant by “success rates” if the goal is query efficiency?  It’s defined later, but since the term is used here, it would be good to define it here.
4.  In the descriptions of Algorithms 1 and 2, could you please give examples of the problem description, and what is contained in the action proposal, and world model (in the text…I see that full details are given in the appendix, but at least a partial explanation in the text would be helpful).
5.  Also, in the description of Algorithms 1 it would be helpful to clearly label/define the LLM heuristics and the external planner, and how they fit together.  In the Algorithm 1 diagram, the LLMs are listed as inputs and in the text, we find that the Action Proposal and State Evaluator are the heuristics, but it’s not clear on the diagram.
6.  Also, more detail on what is described in the “world context” would be helpful in the paper.  When the action proposal generates a proposed action set, is it choosing the possible actions from a list of possible actions defined in the world context (or somewhere else)?
7.  The paper makes reference to both the external world model and also another world model internal to the LLM.  The algorithm diagrams both refer to world models, but it’s not clear if these are the internal or external world models. (The text refers to “Internal” and “true” world models; it would be helpful to make the algorithm descriptions consistent with these definitions)
8.  Boomerang has fewer LLM calls but higher token usage than ToI.  Is there a way to calculate actual computational cost from the LLM calls and token usage?
9.  With Boomerang, are there problems with context size as the number of queries grows?
10.	Very minor: It would be helpful to define LPG on 373.

**Details Of Ethics Concerns:**

I don't know if this matters but this paper is not really blind.  It was very easy from the writing to tell who the authors are.  They repeatedly refer to their own prior work with named references.  (I don’t know the authors so do not have a conflict of interest in reviewing their work).

---

### Official Review · Reviewer_mWcy · 2024-11-04

**Soundness:** 3
**Presentation:** 2
**Contribution:** 1
**Rating:** 3
**Confidence:** 3

**Summary:**

The paper discusses using large language models (LLMs) for query-efficient planning in complex environments without excessively querying the world model. The authors propose two frameworks: one where LLMs act as heuristics to guide traditional search-based planners, and another where LLMs generate complete plans and adapt based on feedback. Experiments show that using LLMs as generative planners is more efficient in terms of the number of queries needed to reach goals.

**Strengths:**

1. The authors provide a well-structured comparison of various planning approaches based on their success rates in achieving goals within a limited number of world model queries.
2. The analysis includes a comprehensive evaluation of LLM calls and token usage (Table 1), showcasing the efficiency of different methods. For instance, Boomerang requires significantly fewer LLM calls compared to ToI-BFS and ReAct, indicating its effectiveness in balancing query efficiency and computational resource utilization.

**Weaknesses:**

1. Methodologically the authors are not proposing anything new. Several works have already proposed the two methodologies explored by authors. (eg., [1])
2. The generative planning approach with LLMs may be limited when scaling to more complex or longer planning horizons, as the model might forget early feedback or re-propose failed actions over extended sequences.
3. The limitations mentioned by the authors in Section 7 are significant.

**Questions:**

1. (Comment) Ref Figure 3: It is expected that the ToI method requires a higher number of queries to the world model for most problem instances. On a positive note, it compensates by using fewer tokens compared to similar approaches like ReAct.
2. What exactly is the novel contribution that the authors are proposing in this work compared to existing approaches?

**References:**

[1] Kambhampati, S., Valmeekam, K., Guan, L., Verma, M., Stechly, K., Bhambri, S., Saldyt, L. and Murthy, A., 2024. LLMs can't plan, but can help planning in LLM-modulo frameworks. arXiv preprint arXiv:2402.01817.

---

### Official Review · Reviewer_hKCE · 2024-11-04

**Soundness:** 2
**Presentation:** 2
**Contribution:** 2
**Rating:** 3
**Confidence:** 3

**Summary:**

This paper focuses on how agents efficiently access world models for planning in complex tasks. Specifically, it examines and studies two methods of utilizing large language models (LLMs) for planning: one approach treats LLMs as a heuristic for each step in traditional search methods; the other approach uses LLMs directly as planners for the entire action trajectory.

The paper first introduces a framework for query-efficient planning using LLMs; then investigates and extends the existing ReAct method, proposing two new approaches: Tree of Interaction (ToI), which uses the LLM as a heuristic, and Boomerang, which employs the LLM as a generative planner. It discusses connections to existing theories on query-efficient planning algorithms and presents evaluations and ablations on the Robotouille and PDDL planning benchmarks, demonstrating the advantages of the proposed method Boomerang.

**Strengths:**

1. Utilizing LLMs for efficient planning is an important and highly timely topic;
2. The paper attempts to connect the proposed methods with past theories;
3. Multiple experiments validate the effectiveness of the proposed methods for these tasks;

**Weaknesses:**

1. Regarding the proposed framework, it appears to be somewhat informally defined. Is the task defined as an MDP or a POMDP? How is the keyword "query-efficient" emphasized throughout the framework? Is it reasonable to assume a deterministic world model?
2. Regarding the two proposed methods, there are many variable factors in their design, such as whether ToI-BFS needs to traverse every state at each step, whether there is feedback from the world model at each step, and the type of feedback provided. This makes it difficult to draw reliable conclusions from a direct comparison of the results of these two methods. Additionally, drawing definitive conclusions for these two types of methods seems unreasonable; it would be more appropriate to specify under what conditions each method has advantages.
3. The explanation of the core finding, "a LLM planner is more adaptive to feedback from the world model than a traditional planner using a LLM merely as a heuristic," does not seem to be well substantiated in the experimental section of the paper.

**Questions:**

Please refer to the weaknesses section above.

**Details Of Ethics Concerns:**

No concern.

---

### Note · Authors · 2024-11-22

I have read and agree with the venue's withdrawal policy on behalf of myself and my co-authors.